# Life Cycle Assessment (LCA) of Cross-Laminated Timber (CLT) Produced in Western Washington: The Role of Logistics and Wood Species Mix

**Cindy X. Chen, Francesca Pierobon and Indroneil Ganguly \***

Center for International Trade in Forest Products (CINTRAFOR), School of Environmental and Forest Sciences, University of Washington, Seattle, WA 98195, USA; cxchen11@uw.edu (C.X.C.); pierobon@uw.edu (F.P.)
\* Correspondence: indro@uw.edu

**Abstract:** The use of cross-laminated timber (CLT), as an environmentally sustainable building material, has generated significant interest among the wood products industry, architects and policy makers in Washington State. However, the environmental impacts of CLT panels can vary significantly depending on material logistics and wood species mix. This study developed a regionally specific cradle-to-gate life cycle assessment of CLT produced in western Washington. Specifically, this study focused on transportation logistics, mill location, and relevant wood species mixes to provide a comparative analysis for CLT produced in the region. For this study, five sawmills (potential lamstock suppliers) in western Washington were selected along with two hypothetical CLT mills. The results show that the location of lumber suppliers, in reference to the CLT manufacturing facilities, and the wood species mix are important factors in determining the total environmental impacts of the CLT production. Additionally, changing wood species used for lumber from a heavier species such as Douglas-fir (*Pseudotsuga menziesii*) to a lighter species such as Sitka spruce (*Picea sitchensis*) could generate significant reduction in the global warming potential (GWP) of CLT. Given the size and location of the CLT manufacturing facilities, the mills can achieve up to 14% reduction in the overall GWP of the CLT panels by sourcing the lumber locally and using lighter wood species.

**Keywords:** life cycle assessment; mass timber; engineered wood products; logistics; manufacturing; climate change; wood products in Pacific Northwest

---

## 1. Introduction

Originally developed in Austria in the early 1990s, cross-laminated timber (CLT) is a type of large-scale and lightweight-engineered wood product that is commonly used for walls, floors, and roofs in residential and commercial buildings. CLT consists of several layers of lumber boards that are stacked and pressed together in alternating directions to form a solid panel. The typical sizes of CLT panels usually range from 0.6 to 3 m wide, up to 18 m in length, and up to 0.5 m thick.

Compared to traditional wood-based construction materials, CLT is conditionally fire resistance according to experimental fire resistance testing and charring rate studies [1–3]. The fire-resistant ability of CLT can be achieved through "charring", which is when a charred layer is formed during heat exposure and serves as an insulation to protect the remaining structure of the panel. Moreover, CLT has many advantages over other wood materials in terms of physical and mechanical properties. Compared to plywood, CLT is less prone to deformations because the alternate layers of lumber make it strong in both the grain and the perpendicular direction. CLT also has greater consistency compared to solid wood since it is made of layers of lumber that are uniform in size and shape.

Compared to traditional "mass timber" products such as laminated veneer lumber (LVL) and glulam, which are commonly used as headers and beams, CLT may be used as a complete wall or

flooring element in construction. As traditional wood materials may only be small components in contemporary houses and buildings, CLT can be used as the primary material for mid to high-rise buildings, which makes CLT a suitable alternative that can replace concrete and steel in many construction projects.

A compelling characteristic of CLT, compared to other wood-based materials, is that it can be manufactured using small-diameter trees that are considered to have low or no commercial value. Washington forests contain many small-diameter trees because of low commercial value and lack of budget to clear them, which become potential hazards as they are vulnerable to wildfires and pest outbreaks. Finding uses for small-diameter trees can be beneficial in maintaining a healthier forest habitat [4]. Incorporating CLT manufacturing in Washington would open the possibility of using these trees as raw materials and therefore reduce natural disasters and bring benefits to ecosystem protection.

Washington has rich forest resources and a well-developed timber industry. Based on the historical data of lumber production, lumber produced in Washington's sawmills is more than enough to supply a normal scale CLT mill. However, the location of these sawmills can directly affect factors such as travel distance and wood species used for production.

The supply chain for wood products is composed of a network of forest activities, harvest, processing, and distribution [5], and each stage within the supply chain can contribute to the total environmental impact. Due to the idiosyncratic characteristics of CLT, the impacts resulting from the supply chain may be different from other wood products. For example, larger CLT panels may have a length of up to 18 m and a width of up to 3 m, which may require special arrangements such as pilot vehicles during the transportation of CLT panels in countries with strict commercial truck dimension/load standards [6]. However, the truck weight and dimension standards are less restrictive in the U.S., and a pilot vehicle may not be necessary for regular CLT transportation in the U.S. [7,8].

Transportation is an important factor associated with forestry operations and wood products according to economic and environmental variability studies [9–13]. Transportation can post various levels of impact depending on the geographical features of the harvest locations and the operational factors. Since CLT is a relatively new product in the U.S., research associated with its environmental impacts is limited. Many studies have emphasized on the usage stage of wood buildings and their impacts on carbon balance [14–18], whereas the association between material source and site-specific CLT panel transportation is not well understood, especially in the U.S. For example, Liu et al. [16] evaluates the carbon emission of CLT buildings in China and assumes the total transportation distance between facilities to be constant, but the variability associated with transportation is not considered. The impact of CLT production may vary depending on facility availability and raw material. The results of existing studies may not be applicable to the U.S. because of different wood species and mode of transportation used [14–16]. Transportation also plays an important role in evaluating the embodied carbon of wood buildings. Obtaining the materials from a reasonably close source is a premise for reducing the embodied carbon and emissions of the building [19–21].

*Study Need and Objectives*

CLT has been gaining recognition in the U.S. over the years and the state of Washington has expressed interest in developing CLT manufacturing in the state. Recently, a "hybrid CLT building" study estimated the potential use of CLT in various applications for mid-to-high rise buildings in the Pacific Northwest [22]. This study projected that, by 2035, the region can experience an annual demand of 6.6 million cubic feet (or 187,000 cubic meters) of CLT panels, just for mid-to-high rise building constructions. Another project funded by the USFS wood innovations grant undertook a comprehensive supply chain study on CLT production in the Pacific Northwest (PNW) [23], with detailed assessment of material sourcing and various economic assessments. This study provides the much needed environmental perspective of CLT production in the region, by utilizing the techno-economic analysis developed by Brandt et al. [23]. Specifically, this study performed a region-specific cradle-to-gate life cycle assessment (LCA) for CLT production, using data and

technology applicable in western Washington to evaluate the potential environmental impacts of CLT production. In addition, this study focused on various location specific material transportation logistics and different species mixes to provide a nuanced understanding of CLT production in western Washington.

Specifically, the objectives of this study were: (a) to assess the environmental impacts of producing CLT panels in western Washington; and (b) to compare environmental impacts based on different logistics and wood species mix. Hence, this study developed a cradle-to-gate LCA based on existing literature and primary data. Several scenarios with different transportation distances and wood mixes were considered. The study compared the changes in impacts when different parameters were used.

## 2. Materials and Methods

This study used an LCA approach based on the ISO 14040 and ISO 14044 standards [24,25]. LCA is a tool for evaluating the environmental impacts of a product throughout its entire life cycle. A product's life cycle includes raw material extraction, manufacturing and processing, usage, and disposal. The environmental impacts of a production or service system are evaluated based on the inputs and outputs of material and energy occurred at each life cycle stage for a defined functional unit of product. The functional unit is defined as the quantification of the identified function of the product. In this study, a functional unit of 1 $m^3$ of CLT was used.

The material estimates used in this study were drawn from the Techno-Economic Analysis (TEA) for manufacturing cross-laminated lumber [23], which was undertaken with reference to manufacturing CLT in the PNW. A CLT mill with a manufacturing capacity of 52,283 $m^3$ per year (small scale mill) was considered in this study. All the material and energy estimates used in this study were drawn from the aforementioned TEA, including the resin type, the resin volume estimates, and the energy estimates at various stages of the manufacturing process. SimaPro 8 was used to perform the LCA analysis. SimaPro incorporates different LCA databases and impact assessment methods. While most input data were adapted from the TEA, data for processes such as electricity generation, lumber production, and fuel consumption were obtained from the U.S. Life Cycle Inventory database (USLCI). Inventory data for resin production came from a combination of LCI databases and existing literature. The amount of materials required for PUR resin production was obtained from Messmer [26], and the inventory data of the production of these required materials were obtained from the USLCI and the Ecoinvent databases.

### 2.1. System Boundary

The system boundary, shown in Figure 1, started at the forest and ended at the construction site, where the CLT material was delivered. Accordingly, the products and processes factored-in within this LCA analysis included forestry activities, material extraction, manufacturing, transportation, and CLT panels delivered at the building construction site. There were three main stages within the system: forestry activities and resource extraction, lumber production, and CLT production. Impacts associated with building construction, building usage, demolition, and end-of-life were not included in this study. It may also be noted that processes such as the manufacturing of capital equipment, facility maintenance, and labor costs were beyond the scope of this study.

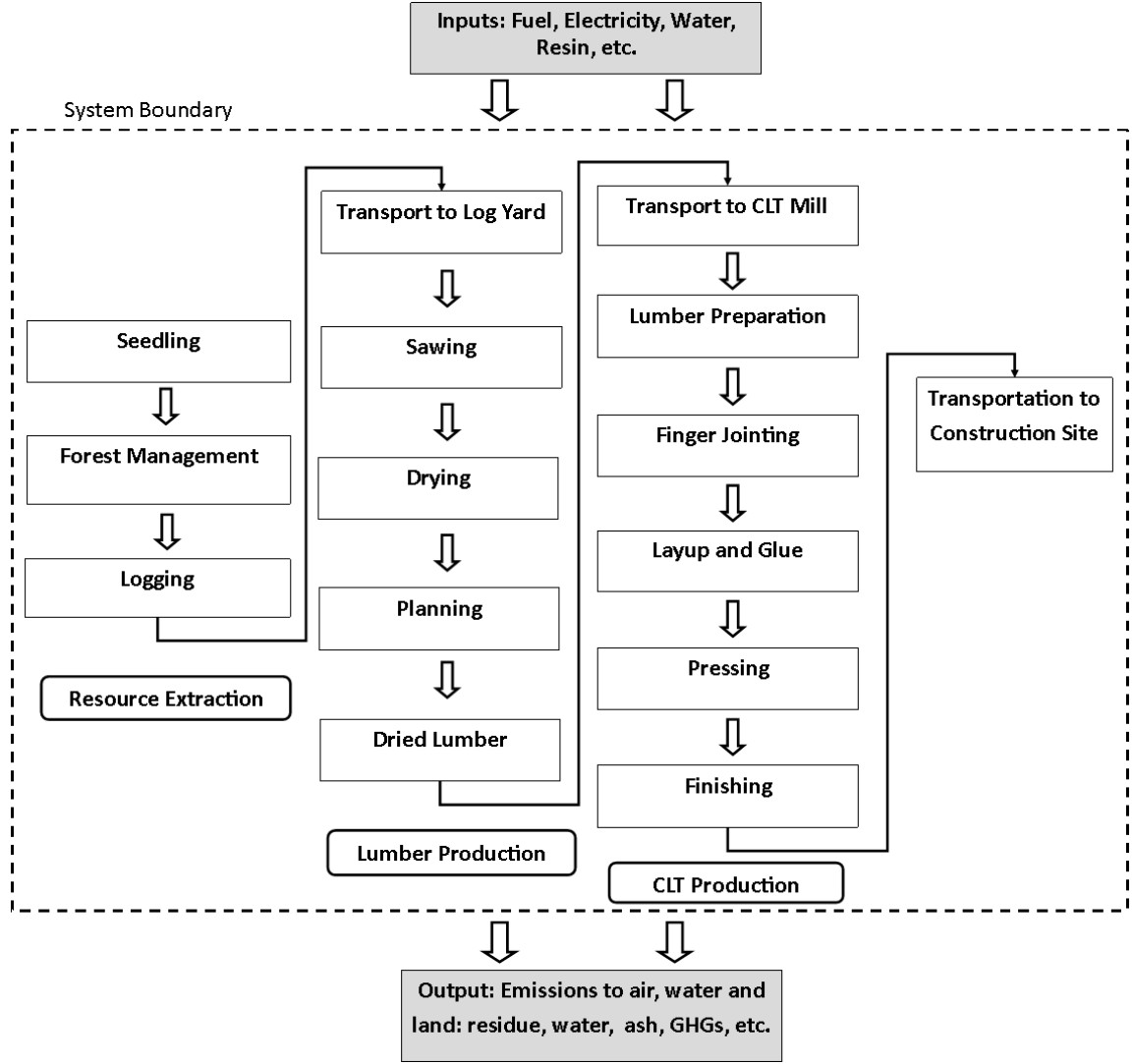

**Figure 1.** System boundary describing the stages involved in the cradle-to-gate LCA.

## 2.2. Assumptions

Several assumptions were made during input calculation and data analysis in terms of material composition, moisture contents, and wood characteristics:

1. For the baseline scenario for CLT production, 50% Douglas-fir and 50% western hemlock, was considered. The bone-dry lumber density for the aforementioned species mix was assumed to be 466 kg/m$^3$, in the baseline scenario [27].
2. Other wood species mixes were considered in the sensitivity analysis: the bone-dry density of lumber constituted of Sitka spruce was assumed to be 360 kg/m$^3$, and that of lumber constituted of Douglas fir was assumed to be 480 kg/m$^3$.
3. Based on the TEA data, the amount of lumber needed to produce 1 m$^3$ of CLT was assumed to be approximately 1.21 m$^3$.
4. The moisture content of CLT panels was assumed to be 12% ± 3%.
5. The construction site where the CLT panels were delivered was assumed to be located in the city of Seattle, WA.

*2.3. Impact Assessment*

The life cycle impact assessment included several indicators that describe the level of impacts generated by a process or system. The Tool for the Reduction and Assessment of Chemical and Other Environmental Impacts (TRACI) was used to model the environmental impact. TRACI is a method developed by the U.S. Environmental Protection Agency (EPA) to estimate the impacts of a specific process system and includes impact indicators at the global, regional, and local levels. As shown in Table 1, the impacts considered in this study included global warming potential (GWP) ($CO_2$ equivalent), acidification potential ($SO_2$ equivalent), photochemical smog potential ($O_3$ equivalent), eutrophication (N equivalent) and stratospheric ozone depletion potential (CFC-11 equivalent). All impact categories were considered midpoint [28–31]. The five impact categories shown in Table 1 are consistent with the categories required for environmental declaration of wood products in North America [32,33].

**Table 1.** Impact categories used to evaluate the environmental impacts of the system.

| Impact Category | Impact Scale | Unit |
| --- | --- | --- |
| Global Warming | Global | kg $CO_2$ equivalent |
| Acidification | Regional/Local | kg $SO_2$ equivalent |
| Photochemical Smog | Local | kg $O_3$ equivalent |
| Eutrophication | Local | kg N equivalent |
| Stratospheric Ozone Depletion | Global | kg CFC-11 equivalent |

The impact on global warming of a process or system may be modeled at different timeframes. The most common timeframe used in current LCAs is 100 years, but 20- and 500-year timeframes may be used as well. In accordance with the practice within the North American wood products LCA [32], in this study, the impact on global warming was calculated using a 100-year timeframe.

*2.4. Lumber Inputs*

The main material for CLT manufacturing is lumber. Lumber may be produced from a wide variety of tree species and has different densities depending on the raw materials mix. In the PNW, raw materials for lumber may include a mix of common tree species such as Douglas-fir, western hemlock, spruce, and other conifers. The minimum specific gravity (SG) requirement of lumber for CLT manufacturing is 0.35 kg/m$^3$ [34,35], but common tree species such as Douglas-fir and western hemlock have SGs that range from 0.45 to 0.48 kg/m$^3$. The bone-dry mass of the wood portion of CLT was about 466 kg/m$^3$ in the baseline scenario. The lumber input requirement for CLT manufacturing was adopted from multiple studies [23,36,37] and 1.21 m$^3$ of lumber was assumed for every m$^3$ of CLT. The LCA for softwood lumber by Milota [27] was used to assess the impacts from lumber production.

*2.5. Transportation*

To compare the impacts associated with transportation logistics, five sawmills in western Washington were selected based on various factors including size, location, capacity, etc. The selected sawmills included Sierra Pacific sawmill operations located in Aberdeen and Burlington, Hampton lumber mill in Darrington, Interfor sawmill in Port Angeles, and Weyerhaeuser located in Raymond. The sawmills are all mid to large scale sawmills in terms of production capacity, which make them plausible suppliers for potential CLT mills in Washington. Two potential CLT mills were hypothetically located near the cities of Forks and Darrington. Both locations have long traditions of lumber production and are looking to develop CLT manufacturing facilities in the area. Both areas are home to timberlands and various tree species, including the major raw materials for softwood lumber such as Douglas-fir (*Pseudotsuga menziesii*), western hemlock (*Tsuga heterophylla*), and Sitka spruce (*Picea sitchensis*). These locations are also viable sites for supplying CLT to major cities across the state.

The construction site was assumed to be located in Seattle, WA. The locations and routes between the facilities are shown in Figure 2.

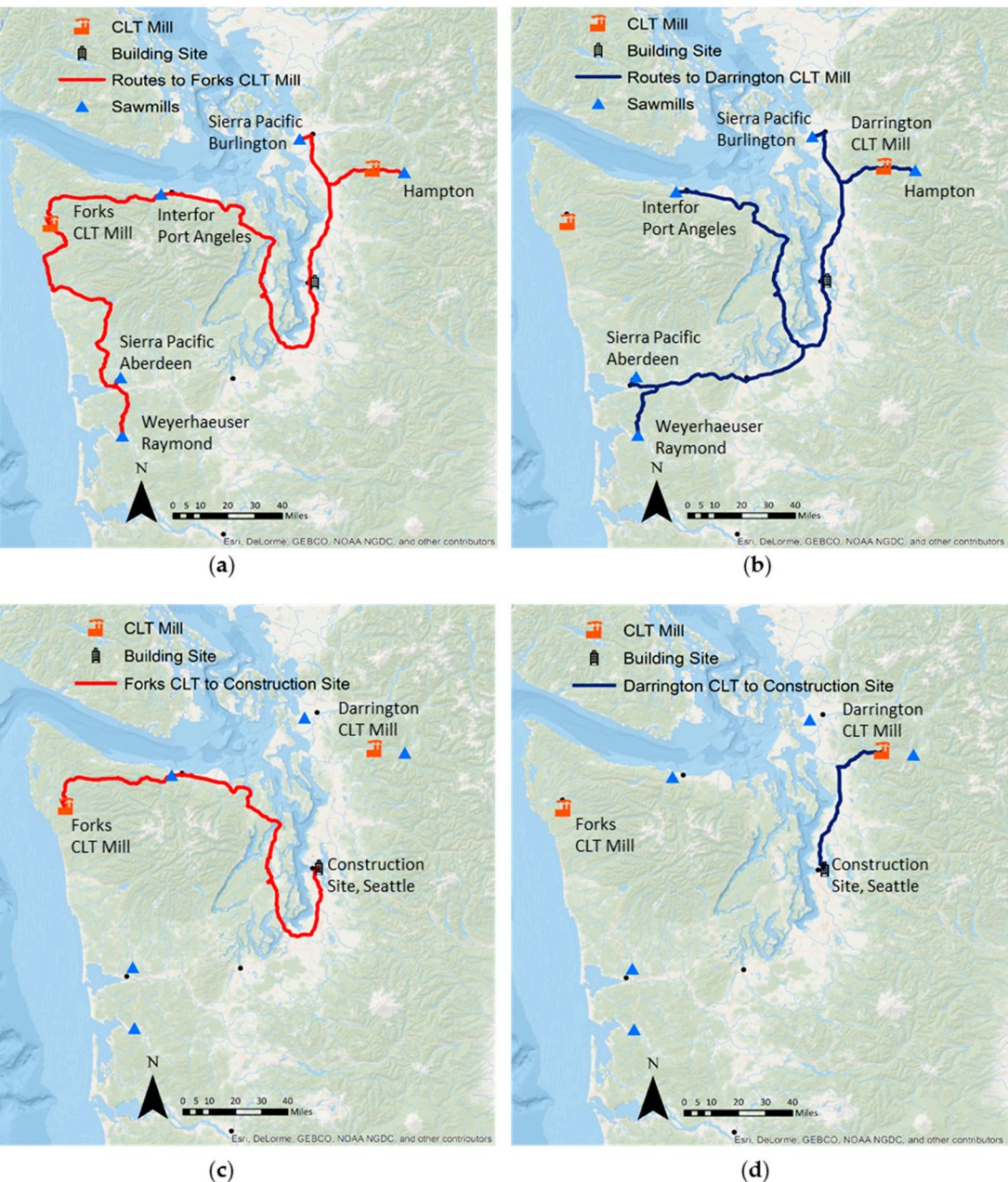

**Figure 2.** Maps of western Washington, indicating the locations and travelling routes of the selected sawmills, hypothetical CLT mills, and construction site: (**a**) transportation routes from sawmills to CLT mill in Forks; (solution) transportation routes from sawmills to CLT mill in Darrington; and (**c,d**) transportation routes from the two hypothetical CLT mills to a building site in Seattle.

Since the transportation conditions of materials between forests, sawmills, CLT mills, and construction sites may vary widely depending on geographical factors, the selected sawmill sites were scattered around different regions across western Washington to make better representations of logistics conditions. Highways are usually located within accessible range for all of the selected sawmills. The location of the sawmills may also affect the wood species used for lumber production.

One key aspect of this study was accounting for the impacts associated with material transportation and considering the effects of transportation distances. Table 2 shows the distances

between the selected sawmills, hypothetical CLT mills, and the final construction site. Combination trucks were used for transporting the lumber from the sawmills to the CLT mills for CLT production. ArcGIS was used to model the transportation distances between the sawmills and the CLT mills. Since larger commercial trucks are needed to transport the lumbers used for CLT manufacturing, certain road restrictions were taken into account. For instance, water transportation such as ferries was strictly avoided and roads that do not allow truck access were also avoided during GIS modeling. The finished CLT panels were then transported to the construction site from the corresponding CLT mills.

**Table 2.** Transportation distances from selected sawmills to potential CLT mills, and from CLT mills to the construction site.

| Transportation of Lumber and CLT | Mode | Distance (km) |
|---|---|---|
| **To Forks CLT Mill** | Truck | |
| Hampton Sawmill | | 440 |
| Interfor Sawmill (Port Angeles) | | 91 |
| Weyerhaeuser (Raymond) | | 206 |
| Sierra Pacific Sawmill (Aberdeen) | | 174 |
| Sierra Pacific Sawmill (Burlington) | | 431 |
| **To Darrington CLT Mill** | Truck | |
| Hampton Sawmill | | 21 |
| Interfor Sawmill (Port Angeles) | | 330 |
| Weyerhaeuser (Raymond) | | 302 |
| Sierra Pacific Sawmill (Aberdeen) | | 284 |
| Sierra Pacific Sawmill (Burlington) | | 76 |
| **To Construction Site** | Truck | |
| Forks CLT Mill | | 322 |
| Darrington CLT Mill | | 104 |

Finally, this study considered differences associated with regionally specific wood species mixes used for CLT production. CLT usually consists of 3, 5, or 7 layers of lumber pressed together in alternate directions. Layers in the same direction as the surface layer (the first layer) are referred to as the parallel layers. Generally, parallel layers use better structural graded lumbers, while the perpendicular layers can use lumbers of lower structural grade. The baseline scenario used lumber produced with a 50–50 mix of Douglas-fir and western hemlock, adapted from Milota [27]. Douglas-fir lumber is known for its superior dimensional stability and is likely to have greater aesthetic appeal as compared to western hemlock. Using structural/visual graded lumber for all the layers of CLT is neither necessary nor efficient to attain specific visual grades (V grades) or elasticity grades (E grades) of the CLT panels. Accordingly, different species (in this case, Douglas-fir, Sitka-Spruce and western hemlock) and various structural and visual grades were considered in this study. Different wood species mixes and their overall impacts in the process are discussed in detail in the sensitivity analysis section.

### 2.6. CLT Manufacturing Inputs

CLT manufacturing involves several key phases, including lumber preparation, finger jointing, layup and adhesive application, pressing, and panel finishing. Multiple steps are involved in each key process during manufacturing and require inputs such as fuel and electricity. For example, lumber preparation involves lumber selection, drying, grouping, cutting, etc. and requires different equipment to kiln dry and cut the lumbers. Depending on the capacity of the CLT mills, the manufacturing processes may vary slightly [35].

In this study, a CLT mill capacity of 52,283 $m^3$ per year was considered. Table 3 shows the materials included in 1 $m^3$ of CLT. The main components in a CLT panel include resin (adhesives) and wood. Approximately 6.44 kg of resin is required for every $m^3$ of CLT. Inputs for CLT manufacturing are calculated based on Brandt et al. [23]. The SG for Douglas-fir is 0.48 on a bone-dry basis, and the

SG for western hemlock is 0.45. Mass allocation was applied in this study, in which approximately 17% of hog fuel was produced in total. The amount of co-products from CLT manufacturing were adapted from Puettmann et al. [37] and values were adjusted based on the amount of production relative to this study.

**Table 3.** Primary components of 1 m$^3$ of CLT, including wood and resin portions.

|  | Unit | Amount |
|---|---|---|
| **Primary Product** |  |  |
| CLT | m$^3$ | 1 |
|  | odkg$^1$ | 472.44 |
| Wood Portion | odkg | 466 |
| Resin | kg | 6.44 |
|  |  |  |
| **Co-Products** |  |  |
| Shavings | odkg | 19.45 |
| Finger Joint Waste | odkg | 5.25 |
| CNC Waste and End Cuts | odkg | 72.06 |

[1] Oven-dry kg. The amount of material was on an oven-dry basis.

Energy Inputs

Electricity and fuel are required for CLT manufacturing. The amount of inputs depends on factors such as technology, efficiency, and available equipment of each mill. Table 4 shows the energy input from each process involved in CLT manufacturing. Resin inputs are required in both the finger jointing and the face bonding processes, while natural gas is required for lumber drying. Electricity is the main energy input for operating the equipment used in the processes. The impacts of producing capital goods and equipment were not included in this study.

**Table 4.** Amount of energy inputs during on-site manufacturing of 1 m$^3$ of CLT.

| Primary Product | Unit | Amount |
|---|---|---|
| CLT | m$^3$ | 1 |
| **Inputs** |  |  |
| Lumber Preparation | kWh | 44 |
|  | m$^3$ of Natural Gas | 2.63 |
| Finger Jointing | kWh | 32 |
|  | kg of Resin | 1.61 |
| Lay Up and Adhesive Application | kWh | 3 |
|  | kg of Resin | 4.83 |
| Pressing | kWh | 18 |
| Panel Finishing | kWh | 31.75 |

## 3. Results

The results of the environmental assessment for the lumber transportation phase and the CLT manufacturing phase are presented in this section.

### 3.1. Impacts of Lumber Transportation

The impacts of transportation strongly depend on the distance and road conditions. As shown in Figure 3, the GWP resulting from transporting 1.21 m$^3$ lumbers to a CLT mill in Forks can range from 6 to 29.16 kg $CO_2$ eq., while the GWP for transporting lumbers to a CLT mill in Darrington can result in 1.36–21.88 kg $CO_2$ eq. Similarly, other impact categories show an increasing trend along with increases in travelling distance (Tables 5 and 6). For example, the smog potential resulting from the transportation of lumbers from the selected sawmills to Forks' CLT mill ranges from 1.15 to 5.57 kg $O_3$ eq., and the

smog potential from the transportation of lumbers from the selected sawmills to Darrington's CLT mill ranges from 0.26 to 4.18 kg $O_3$ eq. Assuming the impacts from lumber production stay constant, there could be up to 95% reduction in GWP associated with lumber transportation when a closely located sawmill is used to supply 100% of the lumber for CLT manufacturing.

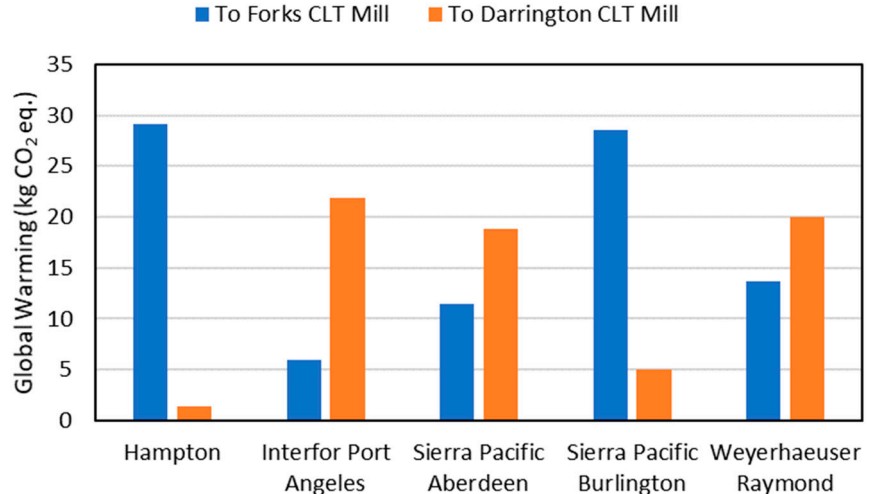

**Figure 3.** Global warming resulting from transporting lumber from selected sawmills to the potential CLT mills.

**Table 5.** Impacts of transporting lumber (lamstock for 1 $m^3$ of CLT) from selected sawmills to CLT mill in Forks, WA.

| Impact Category | Sawmills to Forks CLT Mill | | | | |
|---|---|---|---|---|---|
| | Hampton | Interfor Port Angeles | Sierra Pacific Aberdeen | Sierra Pacific Burlington | Weyerhaeuser Raymond |
| Global Warming (kg $CO_2$ eq.) | 29.16 | 6.0 | 11.5 | 28.55 | 13.62 |
| Acidification (kg $SO_2$ eq.) | 0.21 | 0.042 | 0.081 | 0.2 | 0.096 |
| Eutrophication (kg N eq.) | 0.011 | 0.0024 | 0.0045 | 0.011 | 0.0053 |
| Smog (kg $O_3$ eq.) | 5.57 | 1.15 | 2.19 | 5.45 | 2.6 |
| Ozone Depletion (kg CFC-11 eq.) | $1.2 \times 10^{-9}$ | $2.47 \times 10^{-10}$ | $4.73 \times 10^{-10}$ | $1.18 \times 10^{-9}$ | $5.6 \times 10^{-10}$ |

**Table 6.** Impacts of transporting lumber (lamstock for 1 $m^3$ of CLT) from selected sawmills to CLT mill in Darrington, WA.

| Impact Category | Sawmills to Darrington CLT Mill | | | | |
|---|---|---|---|---|---|
| | Hampton | Interfor Port Angeles | Sierra Pacific Aberdeen | Sierra Pacific Burlington | Weyerhaeuser Raymond |
| Global Warming (kg $CO_2$ eq.) | 1.36 | 21.88 | 18.82 | 5.05 | 20 |
| Acidification (kg $SO_2$ eq.) | 0.0096 | 0.15 | 0.13 | 0.036 | 0.14 |
| Eutrophication (kg N eq.) | 0.00053 | 0.0086 | 0.0074 | 0.002 | 0.0078 |
| Smog (kg $O_3$ eq.) | 0.26 | 4.18 | 3.59 | 0.96 | 3.82 |
| Ozone Depletion (kg CFC-11 eq.) | $5.6 \times 10^{-11}$ | $9.0 \times 10^{-10}$ | $7.75 \times 10^{-10}$ | $2.08 \times 10^{-10}$ | $8.23 \times 10^{-10}$ |

## 3.2. Impacts of CLT Manufacturing

As described in previous sections, the manufacturing of CLT requires multiple processes and each of these contributes to the total impacts. As shown in Table 7, the total GWP for on-site CLT manufacturing is 96.71 kg $CO_2$ eq. per unit of CLT. The resin contributes 29.38 $CO_2$ eq. per unit of CLT, which equates to 30% of the CLT production impacts. After adding the impact from lumber production, the total GWP becomes 155.65 kg $CO_2$ eq. The results shown in Table 7 were calculated using mass allocation. The impact of on-site CLT manufacturing is directly associated with the input from industrial equipment and raw material, as well as the amount of waste generated. Among the

steps involved in CLT manufacturing, panel finishing requires the most energy, followed by pressing, as these steps consume the most electricity input. Panel layup and adhesive application also contribute significantly to the impact of on-site CLT manufacturing because of the significant amount of resin required in this step. Lumber production contributes 58.94 kg $CO_2$ eq. per unit of CLT. Of this, 85% comes from producing lumber, with 15% assigned to forestry operations. Treatment of biogenic carbon is consistent with the IPCC inventory reporting framework. As carbon emissions from biomass combustion are accounted for under the Land Use, Land-Use Change and Forestry (LULUCF) sector, they were not included in energy emissions reporting for the product LCA [38].

**Table 7.** Impacts of lumber production and on-site manufacturing of 1 $m^3$ of CLT in CLT mill.

| Impact Category | Unit | Lumber Production | On-Site CLT Manufacturing | Total (Without Transportation) |
|---|---|---|---|---|
| Global Warming | kg $CO_2$ eq. | 58.94 | 96.71 | 155.65 |
| Acidification | kg $SO_2$ eq. | 0.63 | 0.81 | 1.44 |
| Eutrophication | kg N eq. | 0.02 | 0.09 | 0.11 |
| Smog | kg $O_3$ eq. | 11.19 | 5.98 | 17.17 |
| Ozone Depletion | kg CFC-11 eq. | $6.18 \times 10^{-9}$ | $4.12 \times 10^{-6}$ | $4.13 \times 10^{-6}$ |

In addition to the on-site manufacturing, the transportation of CLT from the CLT mills to the construction site needs to be considered as it can vary by facility location. As shown in Table 8, the impact of transporting CLT from the Darrington CLT mill is lower than transporting CLT from the Forks CLT mill. This is because the construction site was assumed to be in Seattle, which is geographically closer to Darrington than to Forks.

**Table 8.** Impacts of transporting CLT from CLT mills to the construction site (unit: 1 $m^3$ of CLT).

| Impact Category | Unit | From Forks CLT Mill | From Darrington CLT Mill |
|---|---|---|---|
| Global Warming | kg CO2 eq. | 17.9 | 5.76 |
| Acidification | kg SO2 eq. | 0.13 | 0.041 |
| Eutrophication | kg N eq. | 0.007 | 0.0023 |
| Smog | kg O3 eq. | 3.42 | 1.1 |
| Ozone Depletion | kg CFC-11 eq. | $7.37 \times 10^{-10}$ | $2.37 \times 10^{-10}$ |

*3.3. Total Impacts*

The overall environmental impacts from CLT manufacturing include impacts from forestry operations, lumber production, resin production, transportations, and on-site manufacturing. The location of production sites plays an important role in the total impacts. The addition of adhesives is also an important factor to consider. Although both lumber production and CLT manufacturing contribute significantly to the total impacts of CLT, changing the manufacturing process of lumber and CLT in an attempt to reduce environmental impacts is technically challenging and may require significant capital investment. Conversely, changing the sourcing of lamstock is a practical solution to reduce the overall environmental impacts of CLT without negatively affecting its economics. For this reason, this study focused on logistics and analyzed in detail the impact of transportation.

The total impacts for all impact categories are provided in Tables 5–8. Considering GWP, keeping all other processes constant, different contributions from lumber transportation are shown in Figures 4 and 5. The highest GWP impact is from transporting lumbers from the Hampton sawmill to a CLT mill in Forks, reaching over 202 kg $CO_2$ eq., when including impacts of lumber production, transportation, and on-site manufacturing. CLT manufacturing accounts for the highest proportion of the total impacts. In the case of the CLT mill in Forks, the total impact can be reduced by as much as 11.4% if a close-by sawmill is selected. For the CLT mill in Darrington, a reduction of as much as 11.1%

can be achieved. All other impact categories, namely acidification, eutrophication, smog and ozone depletion, show similar trends, and the results are provided in Tables 5–8.

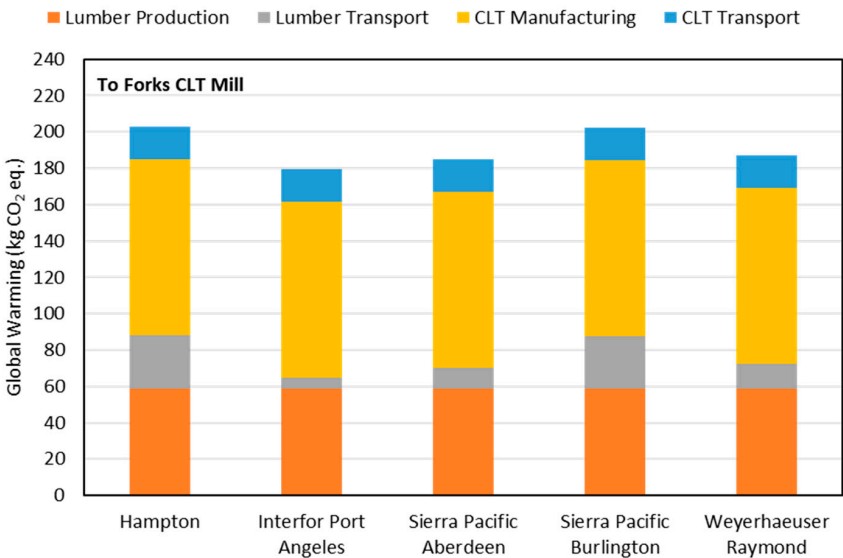

**Figure 4.** Total impacts of 1 m$^3$ of CLT production in Forks, WA, from the forest site to the gate of the construction site.

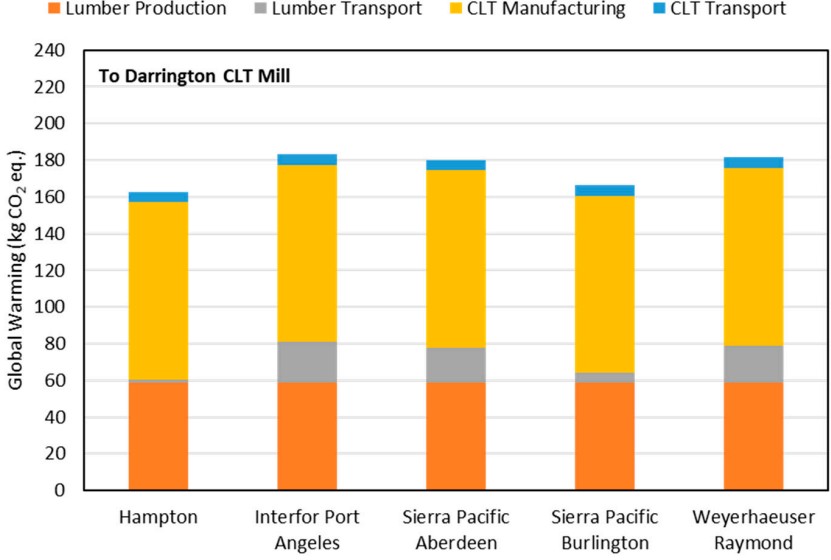

**Figure 5.** Total impacts of 1 m$^3$ of CLT production in Darrington, WA, from the forest site to the gate of the construction site.

### 3.4. Sensitivity Analysis: Wood Species Mix

In addition to the transportation distance, many other factors may influence the total impacts of CLT production. One key factor is the relative density of materials being transported. The baseline scenario of this study assumed that the raw material for lumber consists of Douglas-fir and western hemlock. However, there are many possible wood mixes for producing lumber and CLT, some of which are lighter materials and some are heavier depending on the SG of the wood species. According to the American National Standard for CLT, there are several different grading standards for CLT: E1, E2, E3, E4, V1, V2, and V3. For example, V1 grade CLT uses Douglas-fir–Larch lumbers on all layers, which is heavier due to higher SG of Douglas-fir (0.48) and western Larch (0.52), while CLT using spruce–pine–fir is significantly lighter (SGs of 0.35–0.4).

A sensitivity analysis was performed to determine the changes in total impacts when different wood mixes were used. This section tests the difference of impacts when using Douglas-fir only (heavier) and using Sitka spruce only (lighter), both very common wood species in the PNW, for CLT production. Sitka spruce is a common species in western Washington and meets the minimum requirement for structural lumber, making it a viable material for CLT. Douglas-fir has a SG of 0.48, while Sitka spruce has a SG of 0.36 [39]. In other words, the bone-dry mass of 1 m$^3$ of lumber would be 480 kg and 360 kg, respectively. The mass of the lumber required for CLT production was adjusted according to the SG of the new wood species mix. While the transportation distances between facilities remained constant, the input data for the LCA model changed because of the different SG, which, in turn, changed the environmental impacts.

Figures 6 and 7 show the differences in GWP from transporting lumbers made of a Douglas-fir and western hemlock mix (baseline case), Douglas-fir only, and Sitka spruce only. Because Sitka spruce is noticeably lighter than Douglas-fir and western hemlock, there is a clear declining trend in GWP under all transportation cases. For instance, the impact for transporting spruce-only lumber from Hampton sawmill to Forks' CLT mill shows a 29% decrease compared to the baseline case. On the other hand, the impact for using lumber made of Douglas-fir only shows a slight increase compared to the baseline case, but the difference is relatively small. For example, there is a 3% increase in GWP when transporting lumbers made of Douglas-fir only. This is because the SG of Douglas-fir is only slightly higher than that of the Douglas-fir and western hemlock mix. The same decreasing trends are observed for other impact categories: for instance, there is a 3% increase in acidification potential when transporting Douglas-fir-only lumbers compared to the baseline scenario, same as the GWP.

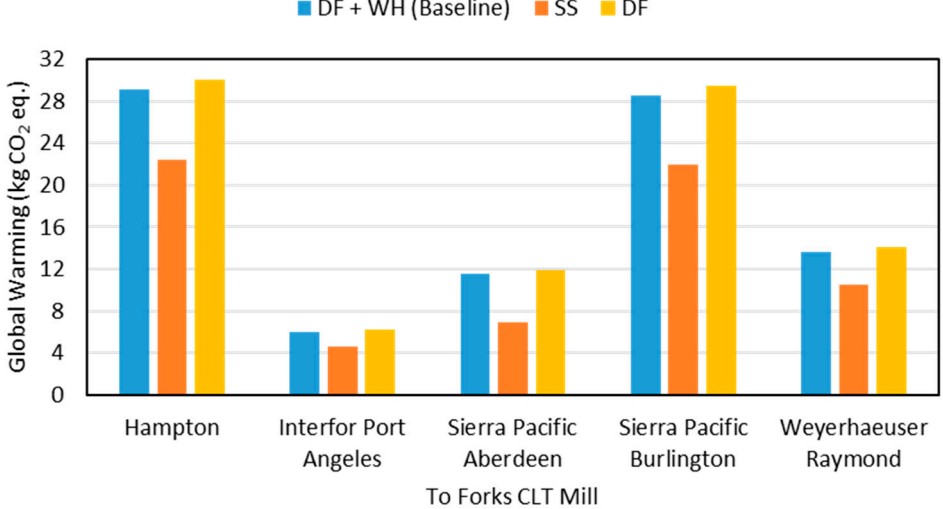

**Figure 6.** Impact on global warming of transporting lumber of different wood species from sawmills to CLT mill in Forks, WA. DF represents Douglas-fir, WH represents western hemlock, and SS represents Sitka spruce. The baseline scenario was a 50–50 mix of Douglas-fir and western hemlock (unit: 1.21 m$^3$ of lumber).

Changes in the type of wood species used can directly influence the choice of the lumber supplier. In this case, although Sitka spruce is a common wood species in western Washington, not all sawmills produce spruce-only lumbers. Sitka spruce is mainly distributed along the coastal areas of PNW and the Puget Sound region. Out of the five selected sawmills, Hampton and Interfor indicate that they supply spruce-pine fir (SPF) lumbers. On the other hand, most of the sawmills in western Washington produce lumbers using Douglas-fir since it is distributed in all forests across the state, which makes Douglas-fir a viable species in all five sawmills. The changes in total GWP as a result of different wood species use are shown in Table 9. By using lighter materials for lumber, the reduction in total GWP can compensate for the impacts of transportation. This is particularly important for sawmills located

further away from the CLT mills. Hampton sawmill is located the furthest away from Forks' CLT mill, resulting in a total GWP of 185.69 kg $CO_2$ eq. for 1 m$^3$ of CLT under the baseline scenario, where a mix of Douglas-fir and western hemlock was used. If Sitka spruce were used as the primary species for lumber production, the total GWP would be reduced to 178.11 kg $CO_2$ eq. On the other hand, Interfor sawmill is located close to Forks, and the reduction in GWP for using Sitka spruce instead of a heavier wood species mix is not as high as compared to that of Hampton. A noticeable reduction for using Sitka spruce occurs if Interfor sawmill supplies lumber to the CLT mill in Darrington, since the two facilities are further away from each other. In general, in the scenarios where Douglas-fir was used as the primary species, the results show an increase in the total GWP across all sawmills given its high specific gravity.

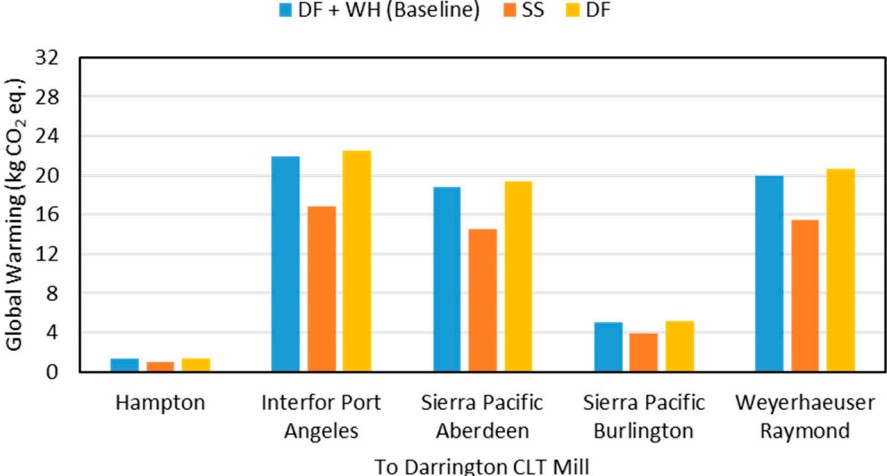

**Figure 7.** Impact on global warming of transporting lumber of different wood species from sawmills to CLT mill in Darrington, WA. DF represents Douglas-fir, WH represents western hemlock, and SS represents Sitka spruce. The baseline scenario was a 50–50 mix of Douglas-fir and western hemlock (unit 1.21 m$^3$ of lumber).

**Table 9.** Comparison of total GWP of CLT production under different case scenarios based on wood species and travelling distances, excluding transportation of CLT to construction site (unit: 1 m$^3$ of CLT).

| Total Global Warming ($CO_2$ eq.) | Douglas-fir/Western Hemlock | Sitka Spruce | Douglas-fir |
|---|---|---|---|
| **For Forks CLT Mill** | | | |
| Hampton | 184.81 | 178.11 | 185.69 |
| Interfor (Port Angeles) | 161.65 | 160.28 | 161.83 |
| Sierra Pacific (Aberdeen) | 167.15 | NA | 167.49 |
| Sierra Pacific (Burlington) | 184.21 | NA | 185.06 |
| Weyerhaeuser (Raymond) | 169.28 | NA | 169.68 |
| **For Darrington CLT Mill** | | | |
| Hampton | 157.02 | 156.7 | 157.06 |
| Interfor (Port Angeles) | 177.54 | 172.5 | 178.2 |
| Sierra Pacific (Aberdeen) | 174.48 | NA | 175.05 |
| Sierra Pacific (Burlington) | 160.7 | NA | 160.85 |
| Weyerhaeuser (Raymond) | 175.65 | NA | 176.26 |

Factoring in CLT mill location, raw material procurement site and wood species mix, we can state that CLT mills can achieve up to 14% reduction in the overall GWP of the CLT panels by sourcing the lumber locally and using lighter wood species. For instance, the total GWP of the CLT panels at the Forks CLT mill, produced out of Sitka spruce procured from the Interfor sawmill, is 160.28 $CO_2$ eq./m$^3$, as compared to that of the CLT panels, produced at the same site, out of Douglas-fir procured from the

Hampton sawmill, which is 185.69 $CO_2$ eq./m$^3$ (Table 9). Similarly, the Darrington CLT mill can reduce the GWP from 178.2 $CO_2$ eq./m$^3$ for CLT panels made with Douglas-fir procured from the Interfor mill to 156.7 $CO_2$ eq./m$^3$, if they produce CLT out of Sitka spruce procured from the Hampton mill.

## 4. Discussion

The results of this study demonstrate that factors such as location of sawmills, road condition, and wood species mix can play an important role in influencing the total impacts associated with CLT manufacturing. When considering the transportation of CLT, it is important to investigate the appropriate routes for trucks, as well as the geographical condition of the region. Although the direct distance between facilities may appear shorter, they are sometimes divided by regional geographical features (i.e., lakes, rivers, mountains, etc.), which lead to longer transportation routes. Among the five sawmills studied, Hampton sawmill is the closest to the hypothetical CLT mill in Darrington, while the Interfor sawmill in Port Angeles is the closest to the hypothetical CLT mill in Forks. On the other hand, these two sawmills also have the furthest travelling distances to the other CLT mill. This is because the two CLT mills are divided by Puget Sound: Forks CLT mill is on the west of Puget Sound, while the Darrington CLT mill is on the east of Puget Sound. To get to the CLT mill in Forks from Hampton, the vehicle needs to go around the Puget Sound instead of going in a relatively direct route. This adds time and distance to the travel, which, in turn, increases the transportation impacts. Although the linear distance between the Hampton sawmill and Forks' CLT mill may not be the furthest, the actual route condition lead to higher impacts for transporting the same amount of lumber from this sawmill to the CLT mill, as compared to the transportation from other sawmills. Same considerations are valid for the transportation between the Interfor sawmill and the CLT mill in Darrington.

Transportation distance is not the only factor that determines the total impacts of CLT manufacturing. It is important to take into account the type of raw materials used for production. For instance, the GWP for transporting lumber made of a mix of Douglas-fir and western hemlock from Hampton sawmill to the CLT mill in Forks is approximately 29 kg $CO_2$ eq., but only 12.5 kg $CO_2$ eq. for transporting the same amount of Sitka spruce lumber. In some cases, replacing heavier lumber with lighter lumber offsets the impacts posted by longer transportation distance for sawmills located further away. For example, under the baseline scenario where a mix of Douglas-fir and western hemlock is used, the highest impact is produced when using lumbers from Interfor sawmill in Port Angeles for CLT manufacturing in Darrington, in which the GWP can reach 183.3 kg $CO_2$ eq. per unit of CLT, which is the highest among all five sawmills. If Sitka spruce were used as the primary species for lumber, the total GWP would be 176.94 kg $CO_2$ eq., lower than two of the sawmills that use Douglas-fir and western hemlock mix. The results of this study suggest that considering the availability of lighter wood species around a lamstock supplier is important from an environmental perspective. Lighter wood species mixes are likely to significantly reduce the overall environmental impacts of CLT production.

The results of this study are consistent with the existing literature. An LCA study of CLT panels produced in Canada indicates significantly lower environmental impacts, as compared to this study [36]. The Canadian study uses a spruce–pine–fir (SPF) mix as lamstock, which is a much lighter wood species mix (~417 kg/m$^3$). It also reports a significantly lower amount of resin use as compared to the study on CLT production in the U.S. [37]. Further, this study assumed that 100% of the lamstock comes from a single sawmill, whereas, in practice, it is possible to obtain lamstock from different sawmills. It is worth noting that the impacts of different CLT mill capacities were not considered in this study. Changes in mill capacity may lead to further impact variations. Similar to the economies of scale, the production process may become more environmentally efficient on a per-unit basis with an increase in the production capacity.

## 5. Conclusions

This study developed a regionally specific cradle-to-gate LCA for CLT production in western Washington and provided data associated with the potential environmental impacts of establishing

CLT mills in two hypothetical locations in the state. The environmental impacts are closely associated with factors such as transportation and wood species mix. Any change in these parameters can influence the environmental impact estimates of CLT production. CLT manufacturing facilities can achieve up to 14% reduction in the overall GWP of the CLT panels by sourcing the lumber locally and using lighter wood species.

Given lumber production, forestry related activities and wood transportation play an important role in the overall environmental impact of the CLT panels, access to locally available species and the existing regional harvesting practices is critical in the overall environmental assessment of CLT panels. For the purpose of reducing total environmental impacts, CLT manufacturers need to consider the travelling distance and the type of lumber used for production. In most cases, getting lumber from close-by lumber suppliers can reduce the environmental impacts. Nonetheless, if lumber, or lamstock, suppliers are located in a setting where lighter wood species are available, it may be rational to obtain lumber from these suppliers even if they may be further away.

Non-wood raw materials used for manufacturing CLT can be influential in the overall environmental impact assessment of CLT. Specifically, the amount and type of resin may lead to significant impact variations. For instance, when using lower amount of resin in manufacturing, the overall impacts of CLT manufacturing is reduced compared to the results of this study [33]. This study modeled the impacts of CLT based on the use of PUR resin, whereas Puettmann et al. [37] used melamine–formaldehyde (MF) resin. The difference in chemical composition and production process of the resins are potential factors that can change the impacts of CLT production. The results of this study also show that resin use contributes to 30% of the overall CLT production GWP impacts (29.38 $CO_2$ eq./m$^3$ of CLT production) and 15–19% of the overall CLT panel GWP impact. Other studies have indicated that the CLT panel can attain the necessary grades by using significantly lower quantity of resin given the CLT volume [36]. Improving the efficiency of resin production and its use in CLT is a key step in limiting the environmental impact of CLT production, which is beyond the scope this study.

**Author Contributions:** Conceptualization, C.X.C., F.P. and I.G.; software and data analysis, C.X.C. and F.P.; validation, F.P. and I.G.; writing—original draft preparation, C.X.C.; writing—review and editing, C.X.C., F.P. and I.G.; visualization, C.X.C. and F.P.; supervision, I.G.; and project administration and project lead, I.G.

**Funding:** This research was made possible by funding from: (1) the USDA National Institute of Food and Agriculture's grant, project No. WNZ-A110355—Accession No. 1011392; and (2) the USDA National Institute of Food and Agriculture's UW-SEFS McIntire-Stennis grant, project No. WNZ-04162015-GE—Accession No. 1006435.

**Acknowledgments:** The authors acknowledge Kristin Brandt and Michael Wolcott of Washington State University for their contribution in providing material and energy input data associated with CLT manufacturing. The authors also acknowledge Luke Rogers of the University of Washington for giving us access to the GIS data used in the analyses.

**Conflicts of Interest:** The authors declare no conflict of interest.

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
