# Peer review of "Life Cycle Assessment (LCA) of Cross-Laminated Timber (CLT) Produced in Western Washington: The Role of Logistics and Wood Species Mix"

_sustainability, doi:10.3390/su11051278_

Round 1

Reviewer 1 Report

This paper is dedicated to the environmental drawbacks quantification by Life Cycle Assessment of CLT production, with an emphasis of the transportation step. The authors used real primary data from a report (ref 27), which is unfortunately not available yet.

This works seems to be a conscientious job, done by qualified persons, with a real expertise about CTL. However I have the following remarks:

-           The authors do not indicate how is performed the LCA. With which software? With which database for the inventory? As the value of this work is in the application of LCA, this has to be transparent.

-           The paper is focused on the transportation step but in Figure 4 and 5, the main contributors for GW are the lumber production and the CLT manufacturing. The authors should explain why scenarios and sensitivity analysis are not performed to tackle the main steps but the transportation one. 

-           The authors have to discussed in details the impacts of lumber production (how is managed the biogenic carbon in this study ?, what are the contributors of the almost 60 kg CO2eq ?) and of CTL manufacturing (resin contribution?). With the current information, it is impossible to determine the overall quality of the assessment.

-           Only global warming is provided for the whole system (Fig 4 & 5), the authors should give and discuss the others impacts.

-           Why only 5 impacts ?

Minor remarks

232 Table 3 an explanation of the unit “odkg” should be added

Table 5,6,8 Fig 4 to 7 Global warming potential should be replaced by global warming (GWP is the characterization factor, not the result)

Author Response

All line numbers indicated in the responses refer to the corresponding line numbers under the “Simple Markup” or “No Markup” view of the manuscript in MS Word.

Reviewer 1

Major Remarks:

The authors do not indicate how is performed the LCA. With which software? With which database for the inventory? As the value of this work is in the application of LCA, this has to be transparent.

-          A detailed description of the software and inventory data used for this manuscript are added in Lines 120-126. An additional reference used for the resin data was added [26].

The paper is focused on the transportation step but in Figure 4 and 5, the main contributors for GW are the lumber production and the CLT manufacturing. The authors should explain why scenarios and sensitivity analysis are not performed to tackle the main steps but the transportation one.

-          Details are added in Lines 307-312.  The paper emphasizes on the role of logistics in CLT production.  Although both lumber production and CLT manufacturing contribute significantly to the total impacts of CLT, changing the manufacturing process of lumber and CLT in an attempt to reduce environmental impacts is technically challenging and may require significant capital investment. Conversely, changing the sourcing of lamstock is a practical solution to reduce the overall environmental impacts of CLT without negatively affecting its economics. For this reason, this study focuses on logistics and analyses in detail the impact of transportation. 

The authors have to discussed in details the impacts of lumber production (how is managed the biogenic carbon in this study?, what are the contributors of the almost 60 kg CO2eq ?) and of CTL manufacturing (resin contribution?). With the current information, it is impossible to determine the overall quality of the assessment.

-          The resin contribution associated with CLT manufacturing, and the contribution of lumber production have been added in Lines 280-282 and Lines 289-293.

Only global warming is provided for the whole system (Fig 4 & 5), the authors should give and discuss the others impacts.

-          The figures for global warming are provided to illustrate the trend of impact changes associated with facility location.  All other impact categories follow the same trend as that of global warming.  The total impact for all categories (e.g. acidification, ozone depletion, eutrophication, etc.), though not expressed in the form of figures, are provided in Tables 5-8.  An explanation is added in Lines 320-322.

Why only 5 impacts?

-          This is explained in Lines 161-162.  References [32] and [33] are added to the manuscript to support the explanation.

Minor Remarks:

Table 3 an explanation of the unit “odkg” should be added.

-          A footer was added to the table to explain “odkg”.

Table 5,6,8 Fig 4 to 7 Global warming potential should be replaced by global warming (GWP is the characterization factor, not the result)

-          Changes are made accordingly.

Reviewer 2 Report

The aim of this paper is relevant. I consider that the subject of the manuscript is interesting, especially because the work resulting has average impact on the society and the environment. 

I. Introduction: Authors should include updates and differences in their work compared to other works on this topic. Especially they should include differences with references 14-18.

II.Materials and MethodsFigure 1 must be improve

III. Conclusions: Authors should reelaborate on the conclusions and compare their results with other research in this field.

Author Response

All line numbers indicated in the responses refer to the corresponding line numbers under the “Simple Markup” or “No Markup” view of the manuscript in MS Word.

Reviewer 2

Remarks:

Introduction: Authors should include updates and differences in their work compared to other works on this topic. Especially they should include differences with references 14-18.

-          Comparison with existing studies are added in Lines 78-83.

Materials and Methods: Figure 1 must be improve

-          Figure 1 was improved by enlarging the font and adjusting the alignment of the text boxes.

Conclusions: Authors should reelaborate on the conclusions and compare their results with other research in this field.

-          Additional comparisons with existing LCA studies associated with CLT production in North America are added in Lines 437-441.

-          Additional details and editing are made to the conclusion section, in particular, the potential impacts of non-wood raw materials (Lines 469-474).

Reviewer 3 Report

The present study evaluates through life cycle assessment the use of cross-laminated timber (CLT), as an environmentally sustainable building material. According to the results, the location of lumber suppliers, and the wood species mix are important factors influencing the total impacts of the CLT production. Additionally, changing wood species used for lumber from a heavier species to a lighter species could generate significant reduction in global warming potential of CLT.

1.      The image quality of Figure 1 should be improved since it is hard to read the text.

2.      What was the LCA-software used for the simulations?

3.      A more detailed description of the steps followed could be provided in the sensitivity analysis.

4.   The discussion should provide more clear conclusions. Also, the reasons why the authors chose to study the distances and the species mix as influential factors.   

Author Response

All line numbers indicated in the responses refer to the corresponding line numbers under the “Simple Markup” or “No Markup” view of the manuscript in MS Word.

Reviewer 3

Remarks:

The image quality of Figure 1 should be improved since it is hard to read the text.

-          Figure 1 has been modified accordingly.

What was the LCA-software used for the simulations?

-          A detailed description of the software and inventory data used for this manuscript are added in Lines 120-126.

A more detailed description of the steps followed could be provided in the sensitivity analysis.

-          Additional details regarding the steps for the sensitivity analysis are provided in Lines 344-348.

The discussion should provide more clear conclusions. Also, the reasons why the authors chose to study the distances and the species mix as influential factors.  

-          Additional details are added in Lines 307-312 to explain the reasons for choosing transportation and species mix as the parameters in this study.

A summary of the findings is added in Lines 406-409 and Lines 434-436 to support the conclusion

Round 2

Reviewer 1 Report

My comments have been taken into account.